# ZEROTH-ORDER ACTOR-CRITIC

## ABSTRACT

Evolution based zeroth-order optimization methods and policy gradient based first-order methods are two promising alternatives to solve reinforcement learning (RL) problems with complementary advantages. The former work with arbitrary policies, drive state-dependent and temporally-extended exploration, possess robustness-seeking property, but suffer from high sample complexity, while the latter are more sample efficient but restricted to differentiable policies and the learned policies are less robust. We propose Zeroth-Order Actor-Critic algorithm (ZOAC) that unifies these two methods into an on-policy actor-critic architecture to preserve the advantages from both. ZOAC conducts rollouts collection with timestep-wise perturbation in parameter space, first-order policy evaluation (PEV) and zeroth-order policy improvement (PIM) alternately in each iteration. The modified rollouts collection strategy and the introduced critic network help to reduce the variance of zeroth-order gradient estimators and improve the sample efficiency and stability of the algorithm. We evaluate our proposed method using two different types of policies, linear policies and neural networks, on a range of challenging continuous control benchmarks, where ZOAC outperforms zeroth-order and first-order baseline algorithms.

## 1 INTRODUCTION

Reinforcement learning (RL) has achieved great success in a wide range of challenging domains, including video games (Mnih et al., 2015), robotic control (Schulman et al., 2017), autonomous driving (Kendall et al., 2019), etc. The majority of RL methods formulate the environment as Markov decision process (MDP) and leverage the temporal structure to design learning algorithms such as Q-learning and policy gradient (Sutton & Barto, 2018). Actor-critic methods are among the most popular RL algorithms, which usually introduce two function approximators, one for value function estimation (critic) and another for optimal policy approximation (actor), and optimize these two approximators by alternating between policy evaluation (PEV) and policy improvement (PIM). On-policy actor-critic methods, e.g., A3C (Mnih et al., 2016) and PPO (Schulman et al., 2017), often use critics to construct advantage functions and substitute them for the Monte Carlo return used in vanilla policy gradient (Williams, 1992), which significantly reduces the variance of gradient estimation and improve learning speed and stability. Among existing actor-critic algorithms, a common choice is to use deep neural networks as the function approximators and conduct both PEV and PIM using first-order optimization techniques.

An alternative approach for RL, though less popular, is to ignore the underlying MDP structures and regard RL problems as black-box optimization, and to directly search for the optimal policy in a zeroth-order way, i.e., without using the first-order gradient information. Recent researches have shown that zeroth-order optimization (ZOO) methods, e.g., ES (Salimans et al., 2017), ARS (Mania et al., 2018) and GA (Such et al., 2017), are competitive on common RL benchmarks, even when applied to deep neural network with millions of parameters. ZOO has several advantages compared to first-order MDP-based RL methods (Sehnke et al., 2010; Salimans et al., 2017; Such et al., 2017; Lehman et al., 2018; Khadka & Tumer, 2018; Qian & Yu, 2021): (1) ZOO is not restricted to differentiable policies; (2) ZOO perturbs the policy in parameter space rather than in action space, which leads to state-dependent and temporally-extended exploration; (3) Zeroth-order population-based optimization possesses robustness-seeking property and diverse policy behaviors.

Despite these attractive advantages, the main limitation of ZOO is its high sample complexity and high variance of the parameter update process, especially in high-dimensional problems. Recent

researches have proposed various techniques to improve ZOO, e.g., using orthogonal or antithetic sampling methods (Sehnke et al., 2010; Salimans et al., 2017; Choromanski et al., 2018; Mania et al., 2018), identifying a low-dimensional search subspace (Maheswaranathan et al., 2019; Choromanski et al., 2019; Sener & Koltun, 2020), or subtracting a baseline for variance reduction (Sehnke et al., 2010; Grathwohl et al., 2018). One of the major reasons for the sample inefficiency of ZOO is its ignorance of the MDP temporal structures. Many recent researches have tried to combine ZOO and first-order MDP-based RL into hybrid methods, e.g., run evolutionary algorithms in parallel with off-policy RL algorithms and optimize the population of policies with information from both sides (Khadka & Tumer, 2018; Pourchot & Sigaud, 2018; Bodnar et al., 2020), or inject parameter noise into existing RL algorithms for efficient exploration (Plappert et al., 2018; Fortunato et al., 2018). However, existing hybrid methods still conduct first-order gradient-based policy improvement (at least as a part), which reimposes differentiable requirement on the policy.

In this paper, we propose the Zeroth-Order Actor-Critic algorithm (ZOAC), which unifies first-order and zeroth-order RL methods into an actor-critic architecture by conducting first-order PEV to update the critic and zeroth-order PIM to update the actor. In such a way, complementary advantages of both methods are preserved, e.g., wide adaptability to policy parameterization, robustness seeking property, state-dependent and temporally-extended exploration. We modify the rollouts collection strategy from episode-wise perturbation as in traditional zeroth-order methods to timestep-wise perturbation, which results in higher sample efficiency and better exploration. We derive the zeroth-order policy gradient under this setting and point out that a critic network can be introduced to estimate the state-value function and trade-off between bias and variance. We then propose a practical algorithm that utilizes several parallelized rollout workers and alternates between first-order PEV and zeroth-order PIM based on generated experiences in each iteration. We evaluate ZOAC on a range of challenging continuous control benchmarks from OpenAI gym (Brockman et al., 2016), using two different types of policies, linear policies and neural networks. Experiment results show that ZOAC outperforms zeroth-order and first-order baseline algorithms in sample efficiency, final performance, and the robustness of the learned policies. We visualize the polices learned in an environment with sparse and delayed reward, which indicates sufficient exploration driven by parameter noise in ZOAC. Furthermore, we conduct ablation studies to demonstrate the indispensable contribution of the modified rollouts collection strategy and the introduced critic network to ZOAC.

## 2 PRELIMINARIES

### 2.1 FROM POLICY GRADIENT TO ACTOR-CRITIC

In standard MDP-based RL settings, the environment is usually formulated as an MDP defined as $(\mathcal{S}, \mathcal{A}, \mathcal{P}, r)$, where $\mathcal{S}$ is the state space, $\mathcal{A}$ is the action space, $\mathcal{P} : \mathcal{S} \times \mathcal{A} \times \mathcal{S} \to \mathbb{R}$ is the transition probability matrix, $r : \mathcal{S} \times \mathcal{A} \to \mathbb{R}$ is the reward function. The return is defined as the total discounted future reward $G_t = \sum_{i=0}^{\infty} \gamma^i r(s_{t+i}, a_{t+i})$, where $\gamma \in (0, 1)$ is the discounting factor. The behavior of the agent is controlled by a policy $\pi(a|s) : \mathcal{S} \times \mathcal{A} \to [0, 1]$, which maps states to a probability distribution over actions. The state-value function is defined as the expected return under policy $\pi$ starting from a certain state: $V^\pi(s) = \mathbb{E}_{a \sim \pi}\{G_t | s_t = s\}$. The goal of MDP-based RL is to find an optimal policy that maximizes the expectation of state-value function under a certain state distribution. Denoting a policy parameterized with $\theta$ as $\pi_\theta$, the objective function can be written as:

$$J_{\mathrm{PG}}(\theta) = \mathbb{E}_{s \sim d}[V^{\pi_\theta}(s)], \quad d = d_0 \text{ or } d^{\pi_\theta} \tag{1}$$

where $d_0$ is the initial state distribution and $d^{\pi_\theta}$ is the stationary state distribution of Markov chain under policy $\pi_\theta$. Generally, the former is used for episodic tasks with finite horizon and the latter is used for continuing tasks with infinite horizon. For any differentiable policy $\pi_\theta$, and for continuing or episodic tasks, the same form of policy gradient can be derived from the policy gradient theorem (Sutton & Barto, 2018). This vanilla policy gradient given by Williams (1992) is as follows:

$$\nabla_\theta J_{\mathrm{PG}}(\theta) = \mathbb{E}_{s_t \sim d_{\pi_\theta}, a_t \sim \pi_\theta}[G_t \nabla_\theta \log \pi_\theta(a_t|s_t)] \tag{2}$$

Vanilla policy gradient suffers from high variance since it directly uses Monte Carlo return from sampled trajectories. Actor-critic methods improved upon it, which usually introduce a critic network to estimate the value function and serve as a baseline to substitute the expected return $G_t$ with a proper form of advantage function $A_t$, for example, TD residual (Mnih et al., 2016), or generalized advantage estimation (GAE) (Schulman et al., 2015). However, the above policy gradient

based methods can only be applied to differentiable policies, and may be unavailable when a non-differentiable controller needs to be optimized.

## 2.2 EVOLUTION STRATEGIES

Existing ZOO methods focus on episodic RL problems with finite horizon and treat them as black-box optimization. In these cases, the length of trajectories is limited and the discounting factor $\gamma$ is usually set as 1. Evolution strategies (ES) is one of the most popular algorithms of ZOO, which optimizes a Gaussian smoothed objective function:

$$J_{\text{ES}}(\theta) = \mathbb{E}_{\epsilon \sim \mathcal{N}(0,I)} \mathbb{E}_{s \sim d_0}[V^{\pi_{\theta+\sigma\epsilon}}(s)] \tag{3}$$

where $d_0$ is the initial state distribution and $\sigma$ is the standard deviation of the Gaussian noise added to the policy. The zeroth-order gradient can be derived using the log-likelihood ratio trick and the probability density function of Gaussian distribution (Nesterov & Spokoiny, 2017):

$$\nabla_\theta J_{\text{ES}}(\theta) = \frac{1}{\sigma} \mathbb{E}_{\epsilon \sim \mathcal{N}(0,I)} \mathbb{E}_{s \sim d_0}[V^{\pi_{\theta+\sigma\epsilon}}(s)\epsilon] \tag{4}$$

In practice, the expectation over Gaussian distribution can be approximated by sampling $n$ noise samples $\{\epsilon_i\}_{i=1,\ldots,n}$, and the corresponding state value $V^{\pi_{\theta+\sigma\epsilon_i}}$ can be approximated by the episodic return $G_i = \sum_{t=0}^{T} \gamma^t r(s_t, a_t)$ of the sample trajectory of length $T$ collected with policy $\pi_{\theta+\sigma\epsilon_i}$:

$$\nabla_\theta J_{\text{ES}}(\theta) \approx \frac{1}{n\sigma} \sum_{i=1}^{n} G_i \epsilon_i \tag{5}$$

The zeroth-order gradient estimator in Equation (5) only relies on the episodic return of each evaluated random directions, so it is applicable to non-differentiable policies. Besides, each perturbed policy remains deterministic in one trajectory, which leads to state-dependent and temporally-extended exploration. Furthermore, the Gaussian smoothed objective also improves robustness of the learned policies in parameter space.

## 3 ZEROTH-ORDER ACTOR-CRITIC

### 3.1 FROM ES TO ZOAC

In this section, we will derive an improved zeroth-order gradient combining the actor-critic architecture for policy improvement. We start from improving the sample efficiency and stability of ES. Most of the existing ES methods applied to RL optimize a deterministic policy (Salimans et al., 2017; Mania et al., 2018), where the exploration is driven by noise in parameter space. Without loss of generality, we follow them in the following derivations and algorithm design. A deterministic policy parameterized with $\theta$ is denoted as $\pi_\theta : \mathcal{S} \to \mathcal{A}$, which directly maps states to actions.

In ES, the policy is perturbed in parameter space at the beginning of an episode and remains unchanged throughout the trajectories. If a large number of random directions $n$ is evaluated, the sample complexity will increase significantly. However, since the zeroth-order gradient is estimated as the weighted sum of several random directions, it exhibit excessively high variance when $n$ is small (Berahas et al., 2021), which may greatly harm the performance. Therefore, it is essential to trade-off this contradictory between sample efficiency and variance.

To encourage sufficient exploration and low variance while maintaining high sample efficiency, here we consider perturbating the policy at every timestep, i.e., the Gaussian noise $\epsilon$ is sampled identically and independently at every timesteps. We regard it as a stochastic exploration policy $\beta = \pi_{\theta+\sigma\epsilon}$, where $\epsilon \sim \mathcal{N}(0,I)$ is Gaussian noise in parameter space and $\sigma$ is the standard deviation. Our objective is to maximize the expectation of the state-value under stationary distribution $d^\beta$ of the exploration policy $\beta$. We can leverage Bellman equation to estimate the state-value via bootstrapping, in which the one-step reward can be replaced with sampled experiences:

$$J_{\text{ZOAC}}(\theta) = \mathbb{E}_{s_t \sim d_\beta}[V^\beta(s_t)] = \mathbb{E}_{\epsilon \sim \mathcal{N}(0,I)} \mathbb{E}_{s_t \sim d^\beta} \mathbb{E}_{s_{t+1} \sim \mathcal{P}}[r(s_t, \pi_{\theta+\sigma\epsilon}(s_t)) + \gamma V^\beta(s_{t+1})] \tag{6}$$

Since all the contents in the outer expectation $\mathbb{E}_{\epsilon \sim \mathcal{N}(0,I)}[\cdot]$ can be regarded as a function of $\epsilon$, the zeroth-order gradient of this objective function can be derived in exactly the same way as in ES.

Moreover, $V^\beta(s_t)$ can be subtracted as a baseline for variance reduction because of its uncorrelation to $\epsilon$ in the outer expectation and the zero mean property of the Gaussian noise $\epsilon$:

$$\nabla_\theta J_{\text{ZOAC}}(\theta) = \frac{1}{\sigma}\mathbb{E}_{\epsilon\sim\mathcal{N}(0,I)}\mathbb{E}_{s_t\sim d^\beta}\mathbb{E}_{s_{t+1}\sim\mathcal{P}}\{[r(s_t, \pi_{\theta+\sigma\epsilon}(s_t)) + \gamma V^\beta(s_{t+1}) - V^\beta(s_t)]\epsilon\} \quad (7)$$

Compared to ES which uses unbiased but high variance Monte Carlo return to evaluate each perturbed policy, the performance of each random direction here is estimated by one-step TD residual with low variance. In practice, a common approach is to introduce a critic network $V_w(s)$ to estimate the state-value function $V^\beta$, which may lead to high bias in this form of advantage estimation.

To trade-off between bias and variance, we consider extending our derivation further to the case where each perturbed policies (i.e., each sampled random noise) run forward $N$ timesteps instead of one timestep only. Equation (7) can be extended to:

$$\nabla_\theta J_{\text{ZOAC}}(\theta)$$

$$= \frac{1}{\sigma}\mathbb{E}_{\epsilon\sim\mathcal{N}(0,I)}\mathbb{E}_{s_t\sim d^\beta}\mathbb{E}_{\mathcal{P}}\left\{\left[\sum_{i=0}^{N-1}\gamma^i r(s_{t+i}, \pi_{\theta+\sigma\epsilon}(s_{t+i})) + \gamma^N V^\beta(s_{t+N}) - V^\beta(s_t)\right]\epsilon\right\} \quad (8)$$

where $\mathbb{E}_\mathcal{P}$ refers to expectation over $N$-step transition dynamics. Similar to one-step case, the cumulative reward within $N$ step can be estimated from sampled experiences when $s_t$ is the first state of the trajectory fragment collected with a certain perturbed policy $\pi_{\theta+\sigma\epsilon}$. By introducing a critic network and choosing an appropriate length $N$, this $N$-step residual advantage function contributes to achieving a good trade-off between the bias and variance.

### 3.2 ANALYSIS ON VARIANCE OF GRADIENT ESTIMATORS

We analyze the variance of these two types of gradient estimators, ZOAC gradient and ES gradient. The budget of timestep of one trajectory $N \times H$ is identical for both algorithms: in ZOAC, each perturbed policy run forward $N$ steps, and $H$ is the number of sampled random directions in one trajectory; in ES, only one perturbed policy is sampled and run forward $N \times H$ steps. If we denote the accumulative reward obtained within $N \times H$ timesteps in ES as $\hat{V}_{NH}^{\pi_{\theta+\sigma\epsilon}}$ and the $N$-step TD residual in ZOAC as $\hat{A}_N^{\pi_{\theta+\sigma\epsilon}}$. We can then estimate the zeroth-order gradient according to Equation (4) and (8) respectively:

$$\nabla_\theta \hat{J}_{\text{ES}}(\theta) = \frac{1}{n\sigma}\sum_{i=1}^{n}\hat{V}_{NH}^{\pi_{\theta+\sigma\epsilon_i}}\epsilon_i \quad (9)$$

$$\nabla_\theta \hat{J}_{\text{ZOAC}}(\theta) = \frac{1}{nH\sigma}\sum_{i=1}^{nH}\hat{A}_N^{\pi_{\theta+\sigma\epsilon_i}}\epsilon_i \quad (10)$$

We now give the upper bound of variance for these two gradient estimators. Variance is defined as the trace of the convariance matrix of gradient vectors $\text{Var}(\boldsymbol{g}) = \sum_{l=1}^{d}\mathbb{E}[g_l^2] - (\mathbb{E}g_l)^2$, where $\boldsymbol{g} = (g_1, g_2, ..., g_d)^\top$ (Zhao et al., 2011). We can derive the variance bound as follows if both the reward and the critic network output is bounded (detailed derivation is provided in Appendix A.3, vectors are bolded in the appendix for clarity but not in the main text).

**Theorem 1.** *If the reward $|r(s, a)| < \alpha$, the critic network output $|V_w(s)| < \beta$, and $n$ trajectories with length of $N \times H$ timesteps are collected in one iteration, the upper bounds of the variance for gradient estimators (Equation (9) and (10)) are:*

$$\text{Var}[\nabla_\theta \hat{J}_{\text{ES}}(\theta)] \leq \frac{(1-\gamma^{NH})^2\alpha^2 d}{n\sigma^2(1-\gamma)^2} \quad (11)$$

$$\text{Var}[\nabla_\theta \hat{J}_{\text{ZOAC}}(\theta)] \leq \frac{((1-\gamma^N)\alpha + (1-\gamma)(1+\gamma^N)\beta)^2 d}{nH\sigma^2(1-\gamma)^2} \quad (12)$$

We can compare their variance in a more intuitive way: if $N \times H = 1000$, $\gamma = 0.99$, and assume that $\beta \approx \frac{\alpha}{1-\gamma}$, the difference of variance bounds becomes $\text{Var}[\nabla_\theta \hat{J}_{\text{ZOAC}}(\theta)] - \text{Var}(\nabla_\theta \hat{J}_{\text{ES}}(\theta)) \approx (\frac{4}{H} - 1)\frac{10000\alpha^2 d}{n\sigma^2}$, which decreases with $H$ and drops below zero when $4 < H \leq 1000$ (i.e., $N$ is smaller than 250). This suggests that although same amount of data is collected, an appropriate rollout length $N$ can indeed reduce variance of the gradient estimators. Besides, both variance bound are inversely proportional to $n$, which urges us to collect more trajectories.

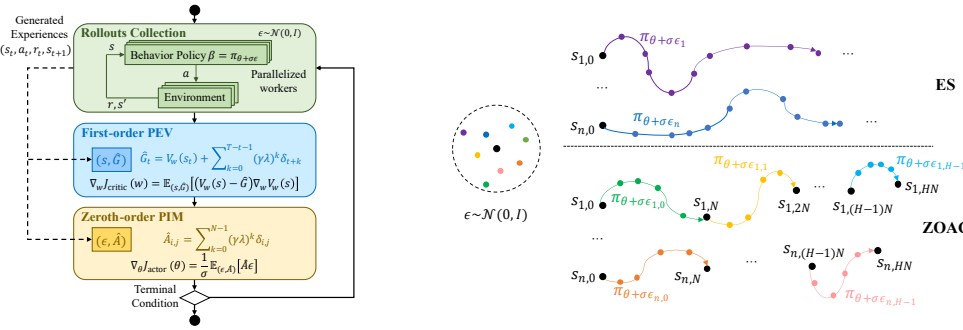

(a) Overall framework of ZOAC     (b) Rollouts collection strategies: ES vs. ZOAC

Figure 1: (a) The overall framework of our proposed algorithm ZOAC; (b) Comparison of rollouts collection strategies (with $n$ parallelized samplers): On the top is ES, which performs episode-wise perturbation; on the bottom is ZOAC, which performs $N$ timestep-wise perturbation.

### 3.3 PRACTICAL ALGORITHM

We propose the Zeroth-Order Actor-Critic (ZOAC) algorithm, which unifies first-order and zeroth-order methods into an on-policy actor-critic architecture by conducting rollouts collection with timestep-wise perturbation in parameter space, first-order policy evaluation (PEV) and zeroth-order policy improvement (PIM) alternately in each iteration. The overall framework of ZOAC is shown in Figure 1a and the pseudocode is summarized in Appendix A.1. In each iteration, parallelized workers will collect rollouts in the environment with perturbed policies, then the agent train the critic network to estimate state-value function under the exploration policy, and finally improve the policy along the zeroth-order gradient direction.

**Rollouts collection.** The rollouts collection strategy is illustrated briefly in Figure 1b, which is a parallelized version with $n$ workers. If we denote the $t$-th state sampled by the $i$-th worker as $s_{i,t}$, the rollout strategy can be described as: when reaching states in $\{s_{i,jN}\}$, where $j \in \mathbb{N}$, a new random direction $\epsilon_{i,j}$ is sampled and the behavior policy is perturbed; when reaching other states, the deterministic behavior policy remains unchanged. It's worth noting that the notation is only for continuing case where an episode is never done. In episodic tasks, the rollout length $1 \le N_{i,j} \le N$ actually varies between different perturbed policies $\pi_{\theta + \sigma \epsilon_{i,j}}$ since an episode may terminate at any time. However, we still use $N$ to denote the rollout length of each perturbed policy for brevity.

A limit case of our proposed strategy is that when $N$ is chosen as the episode length and the critic network is turned off (i.e., $V_w(s) \equiv 0$), the algorithm actually degenerate into ES, since all perturbed policies are evaluated by running a whole episode, and the episodic return is used as the fitness score.

**First-order PEV.** The state-value function $\hat{V}(s)$ can be estimated by a jointly optimized critic network $V_w(s)$, which aims to minimize the MSE loss between the network output and state-value target. In each iteration, in total $n \times N \times H$ states and the corresponding target values $(s, \hat{G})$ are calculated and used for critic training. In a trajectory with length $T$, the target value $\hat{G}_t$ for each state $s_t$ is calculated as (Schulman et al., 2015; Andrychowicz et al., 2021):

$$\hat{G}_t = V_w(s_t) + \sum_{k=0}^{T-t-1} (\gamma\lambda)^k [r_{t+k} + \gamma V_w(s_{t+k+1}) - V_w(s_{t+k})] \tag{13}$$

where $0 < \lambda < 1$ is a hyperparameter to control the trade-off between bias and variance of the value target. In Figure 1a, the one-step TD residual of each state $s$ is denoted as $\delta$ for simplicity. The objective function of PEV can be written as:

$$J_{\text{critic}}(w) = \mathbb{E}_{(s,\hat{G})} \left\{ \frac{1}{2} [V_w(s) - \hat{G}]^2 \right\} \tag{14}$$

In practice, the critic network is constructed as a neural network and updated through several epoches of stochastic gradient descent in each iteration.

Figure 2: Learning curves on MuJoCo benchmarks. Exploration noise are turned off for evaluation. The solid lines correspond to the mean and the shaded regions to the 95% confidence interval over 5 trials using a fixed set of random seeds. All curves are smoothed uniformly for visual clarity.

Table 1: Max total average return within certain environmental steps (mean±std over 5 trials).

| Environment | Inv.D.P.-v2 | Hopper-v2 | HalfCheetah-v2 | Ant-v2 |
|---|---|---|---|---|
| Timesteps | 1e6 | 2e6 | 2e6 | 2e7 |
| **ZOAC(Linear)** | **9359.93±0.01** | **3333.60±97.85** | 5190.82±196.99 | **4495.33±100.37** |
| **ZOAC(Neural)** | 9339.66±5.29 | 3195.15±91.76 | **5339.95±140.46** | 4292.09±108.69 |
| ARS(Linear) | **9359.86±0.06** | 2891.28±305.61 | 2967.98±889.42 | 3427.85±765.62 |
| ES(Neural) | 9155.73±404.25 | 1065.34±49.14 | 2349.11±444.77 | 3274.89±519.66 |
| PPO(Neural) | **9350.89±0.04** | 3178.60±270.09 | 5219.61±677.90 | 3796.13±754.78 |

**Zeroth-order PIM.** We calculate the zeroth-order gradient with $n \times H$ random directions and the corresponding advantage function as $(\epsilon, \hat{A})$. Similar to state value estimation, we leverage the generalized advantage estimation (GAE) trick (Schulman et al., 2015) to further control the bias-variance trade-off. We also perform advatage normalization to ensure consistent gradient length during training. Following the notations in Figure 1b, the advantage function can be written as:

$$\hat{A}_N^{\pi_{\theta+\sigma\epsilon_{i,j}}} = \sum_{k=0}^{N-1} (\gamma\lambda)^k [r_{i,jN+k} + \gamma V_w(s_{i,jN+k+1}) - V_w(s_{i,jN+k})] \qquad (15)$$

where $\lambda$ is the same as in Equation (13). The zeroth-order gradient can be then estimated as the weighted sum of the sampled random directions:

$$\nabla_\theta J_{\text{actor}}(\theta) \approx \frac{1}{nH\sigma} \sum_{i=1}^{n} \sum_{j=0}^{H-1} \hat{A}_N^{\pi_{\theta+\sigma\epsilon_{i,j}}} \epsilon_{i,j} \qquad (16)$$

## 4 EXPERIMENTS

### 4.1 PERFORMANCE EVALUATION

We evaluate the performance of ZOAC on the MuJoCo continuous control benchmarks (Todorov et al., 2012) in OpenAI Gym (Brockman et al., 2016). We choose Evolution Strategies (ES) (Salimans et al., 2017; Liang et al., 2018) and Augmented Random Search (ARS) (Mania et al., 2018) as zeroth-order baselines and proximal policy optimization (PPO) (Schulman et al., 2017; Raffin et al., 2019) as a first-order actor-critic baseline.

We use two different types of policies: linear policies for ARS and ZOAC (linear), neural networks with (64, 64) hidden nodes and tanh nonlinearities for ES, PPO and ZOAC (neural). For a fair comparison, we enable observation normalization for all methods, which has been proved effective no matter in first-order methods or zeroth-order methods (Mania et al., 2018; Andrychowicz et al., 2021). When using neural networks as actors, we also use layer normalization (Ba et al., 2016) in ZOAC and virtual batch normalization in ES (Salimans et al., 2017). Both of them ensure the diversity of behaviors among the population, while the former is less computationally expensive. We summarize the implementation details of ZOAC in Appendix A.2 and follow the recommended hyperparameter settings listed in the related papers or code repositories.

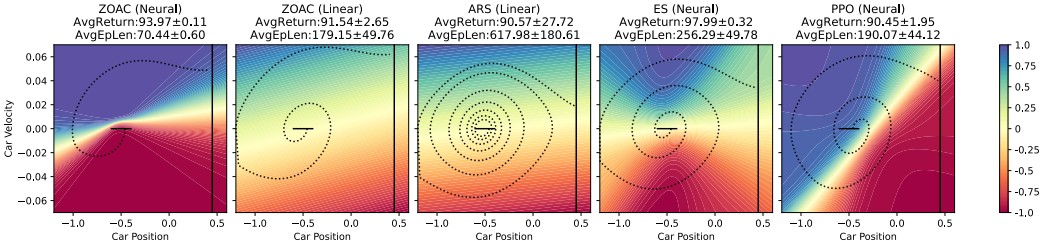

Figure 3: Visualization of the policies learned by different methods on MountainCarContinuous-v0. Color represents the action output, from $-1.0$ (red) to $1.0$ (blue). The horizontal lines in solid indicate the initial car position distribution $x \sim U(-0.6, -0.4)$, and the vertical lines in solid indicate the goal $x > 0.45$. The dashed curves are trajectories starting from the same initial state.

Table 2: Robustness comparison of the learned policies on HalfCheetah-v2. Best performing policies from Figure 2 are tested (4e7 timestep limitation for ARS and ES). Each policy is evaluated on 400 trajectories in the same environment and the average return over 5 policies are listed.

| Noise type | No extra noise | Obs. noise ($\sigma = 0.1$) | Para. noise ($\sigma = 0.05$) |
|---|---|---|---|
| **ZOAC(Linear)** | 4909.53 | 4402.54 (-10.3%) | 3289.15 (-33.0%) |
| **ZOAC(Neural)** | 5179.39 | **5057.07 (-2.4%)** | **4247.08 (-18.0%)** |
| ARS(Linear) | 4529.30 | 1771.23 (-60.9%) | 1724.40 (-61.9%) |
| ES(Neural) | 5478.01 | 4550.22 (-16.9%) | 1527.93 (-72.1%) |
| PPO(Neural) | 4668.55 | 3342.23 (-28.4%) | 972.08 (-79.2%) |

Figure 2 presents the learning curves on four continuous control tasks. Table 1 summarizes max total average return within the timestep threshold over 5 trials. Additional results including state-value estimation and performance comparison of different policies are attached in Appendix A.4 and A.6.

ZOAC matches or outperforms baseline algorithms across tasks in learning speed, final performance, and variance over trials. One thing worth mentioning is that both the zeroth-order baseline methods perform reward shaping to resolve the local optima problem: ARS subtracts the survival bonus from rewards (1 in Hopper and Ant), while ES transforms the episodic returns into rankings. Although these tricks improve the performance, they also alter the update directions of the policies and make it difficult to determine what is the real objective function being optimized. ZOAC, however, surpasses ES and ARS without relying on specific exploration tricks, which can be attributed to the introduction of critic network and the construction of advantage estimations in policy improvement.

**Robustness Comparison.** The objective function of ZOAC aims to maximize the expected state-value of the stochastic behavior policy that contains parameter noise all the time, which intuitively encourages the agent to find a wider optima and leads to better generalization and robustness. Hence, we evaluate the learned policies under two types of noise, observation noise and parameter noise. Extra observation noise is added to the normalized observation at each timestep, which leads to a slightly different observation distribution. Extra parameter noise is added at the beginning of each trajectories, which pushes the learned policy to its neighborhood. The result in HalfCheetah-v2 is presented in Table 2. Results show that in general the policies learned by ZOAC possess higher robustness against both observation noise and parameter noise, which can be ascribed to the robustness-seeking property of our method. The linear policies learned by ARS seem very fragile and suffer significant performance degradation under extra noise, far inferior to the linear ones learned by ZOAC. Additional results and discussions are attached in Appendix A.5.

**Visualization of the learned policies.** In order to intuitively observe the behaviors of the learned policies, we apply all methods on MountainCarContinuous-v0, in which the car is rewarded +100 only when it achieves the goal and penalized by the action output at every timestep. Policy gradient methods usually struggle on this problem because the reward is sparse and delayed, while zeroth-order methods can better handle reward sparsity by nature. We visualize the policies learned by each algorithm in Figure 3. Among these learned policies, the neural policy learned by ES obtains the highest average return, while the neural policy learned by ZOAC obtains the second highest average

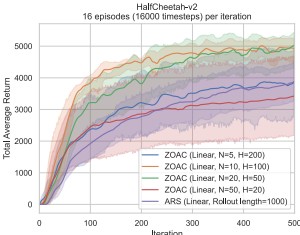 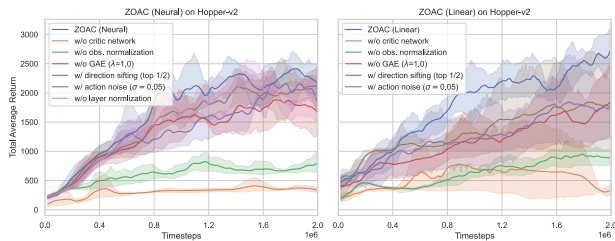

Figure 4: Influence of the rollout length $N$ with each perturbed policies.

Figure 5: Ablation studies on ZOAC with neural network (left) and linear poicy (right).

return within the shortest episode length. The latter has a similar but much steeper terrain compared to the former one and implies a larger control action in most areas. As for linear policies, the one learned by ZOAC also tends to achieve the goal in a shorter episode length than the one learned by ARS. We attribute this to the usage of discounting factor, which pushes the agent to perform higher actions and achieve the goal as early as possible. ZOAC outperforms PPO both in final performance and training stability over different random seeds, due to its state-dependent exploration in parameter space, which is more efficient than action noise and is essential to solve this task.

## 4.2 ABLATION STUDIES

**Appropriate rollout length.** As illustrated in Section 3.2, choosing appropriate rollout length $N$ of each perturbed policies may achieve a good trade-off between bias and variance. We perform an ablation study to understand the effect of timestep-wise perturbation strategy and choice of $N$. We compare ZOAC and ARS on HalfCheetah-v2, the former conducting timstep-wise perturbation and the latter conducting episode-wise perturbation. We use linear policies in both methods and set hyperparameters to the same, including the standard deviation of parameter noise $\sigma$, the learning rate of policy $\alpha_{\text{actor}}$, and also the budget of timesteps within one iteration. Figure 4 shows the influence of $N$ on performance. Results show that under this setting, both 10 and 20 are good choices for $N$ which lead to better performance, while other two choices, 5 and 50, perform similarly to ARS. Note that here we sample as many experiences as ARS per iteration for comparison, while in practice, the number of timesteps collected in each iteration is highly tunable in ZOAC. In fact, we found that rollout length $N$ in a large range, approximately from 5 to 50, perform quite well across tasks.

**Analysis on different components.** To evaluate the contribution of each individual component and also the potential of additional techniques, we perform ablation studies and present the results in Figure 5. Results demonstrate that critic network is a crucial part of ZOAC, i.e., $N$-step accumulative reward without bootstrapping is not sufficient to guide policy improvement. Observation normalization technique is also essential to zeroth-order methods, which helps to generate diverse policies via isotropic Gaussian noise. Besides, GAE trick and layer normalization trick slightly improve the performance. Mania et al. (2018) propose to use only the top performing directions in policy update to relieve the bad influence of noisy evaluation results and validate its effectiveness on ARS. Here we perform a similar direction sifting technique, using only the directions that have the highest advantage in policy improvement, but it seems to pull down the learning performance of ZOAC. Moreover, results show that additional action noise is not helpful to the performance, indicating that the exploration driven by parameter noise is sufficient.

## 5 RELATED WORK

**ZOO and its applications in RL.** At each iteration, ZOO samples several random directions from a certain distribution, and then the distribution is updated according to the evaluation results over these directions. Sehnke et al. (2010) derive parameter-exploring policy gradients (PGPE) for episodic RL problems, which has reduced variance and higher performance than vanilla policy gradient. Salimans et al. (2017) and Such et al. (2017) propose highly scalable evolution strategies (ES) and genetic algorithms (GA) respectively, both of which can be applied to deep neural networks and achieve competitive performance with MDP-based RL algorithms. Mania et al. (2018) propose

augmented random search (ARS), which applied ZOO to linear policies with techniques including observation normalization, reward scaling, top performing directions sifting, and achieve astonishing performance on RL benchmarks considering its simplicity. ZOO has regained popularity in recent years because of its special advantages when applied in RL, including wide adaptability to policy parameterization (e.g., deterministic or stochastic, differentiable or non-differentiable), robustness seeking property, state-dependent and temporally-extended exploration.

**Improved techniques for ZOO.** The main limitation of ZOO is its high sample complexity. Researchers have proposed various improved techniques for ZOO from different perspectives. One way is to adopt advanced Monte Carlo sampling methods to reduce variance of the zeroth-order gradient estimation, e.g., antithetic sampling (Sehnke et al., 2010; Salimans et al., 2017; Mania et al., 2018), orthogonal and Quasi Monte Carlo exploration (Choromanski et al., 2018). Constructing control variates (i.e., subtracting a baseline) is another popular variance reduction technique. Sehnke et al. (2010) adopt a moving-average baseline in PGPE heuristically, while Zhao et al. (2011) derive the optimal baseline for PGPE in an analytical form that minimizes the variance. Moreover, the sample complexity of zeroth-order methods will further increase with the dimension of the optimization problem (Nesterov & Spokoiny, 2017), therefore some researches aim to identify a low-dimensional search space and guide the search towards faster convergence. Guided ES (Maheswaranathan et al., 2019) and ASEBO (Choromanski et al., 2019) are proposed based on a similar idea: to identify linear subspaces and adapt the search distribution from recent history of descent directions. Sener & Koltun (2020) propose LMRS, which uses more expressive neural networks to represent subspaces and jointly learns the underlying subspace and optimizes the objective function.

**Hybridization of ZOO and first-order MDP-based RL.** These two methods have complementary advantages when applied to RL problems, and recent researches have tried to combine them for better performance. Khadka & Tumer (2018) propose the ERL framework that runs evolutionary algorithms (EA) and DDPG (Lillicrap et al., 2015) concurrently with bidirectional information flow, i.e., the DDPG agent is trained with experiences generated by the EA population and reinserted into the population periodically to guide the evolution process. CEM-RL (Pourchot & Sigaud, 2018) and Proximal Distilled ERL (Bodnar et al., 2020) adopt similar hybridization framework, but use different algorithms as components and improve training techniques. Fortunato et al. (2018) and Plappert et al. (2018) inject parameter noises into existing first-order MDP-based RL algorithms to drive more efficient exploration, and demonstrate that existing RL algorithms can indeed benefit from parameter space exploration through comparative experiments. Some other hybrid methods (Grathwohl et al., 2018; Tang et al., 2020) leverage policy gradient and reparameterization trick to construct control variates, which leads to unbiased, low variance gradient estimators. Our proposed method, however, unifies first-order and zeroth-order methods into an on-policy actor-critic architecture by conducting first-order PEV and zeroth-order PIM alternately in each iteration. The state-value function network does not only serve as a baseline to reduce variance, but also as a critic used for bootstrapping, which leads to reduced variance and accelerated learning (Sutton & Barto, 2018). The policy is updated in a zeroth-order way, which implies wide adaptability to different forms of policies.

## 6 CONCLUSION

In this paper, we propose Zeroth-Order Actor-Critic algorithm (ZOAC) that unifies evolution based zeroth-order and policy gradient based first-order methods into an on-policy actor-critic architecture to preserve the advantages from both, including the ability to handle different forms of policies, state-dependent exploration, robustness-seeking property from the former and high sample efficiency from the latter. ZOAC conducts rollouts collection with timestep-wise perturbation in parameter space, first-order policy evaluation (PEV) and zeroth-order policy improvement (PIM) alternately in each iteration. Experimental results in a range of challenging continuous control tasks show that ZOAC outperforms zeroth-order and first-order baselines. Robustness analysis and ablation studies on hyperparameters and components are also performed to show the properties of ZOAC.

Moreover, our methods achieve such improvement while still using traditional isotropic Gaussian noise for perturbation, so in principle those improved techniques for ZOO from sampling perspectives can be further integrated, e.g., Monte Carlo sampling techniques, low-dimensional subspace identification, adaptive perturbation scale, which may lead to even higher performance.

## 7 REPRODUCIBILITY

The code of ZOAC will be released after the author notification in `https://anonymous.4open.science/r/Zeroth-Order-Actor-Critic-1A71`. We summarize the algorithm in Appendix A.1 and describe the implementation details in Appendix A.2.

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

# A APPENDIX

## A.1 PSEUDOCODE OF ZOAC

---
**Algorithm 1** Zeroth-Order Actor-Critic (ZOAC)

---
1: **Initialize:** policy parameters $\theta$, critic network parameters $w$
2: **for** each iteration **do**
3:    **for** each worker $i = 1, 2, ..., n$ **do**
4:       **for** $j = 0, 1, ..., H - 1$ **do**
5:          Sample $\epsilon_{i,j} \sim \mathcal{N}(0, I)$
6:          Run perturbed policy $\pi_{\theta + \sigma \epsilon_{i,j}}$ in environment for $N$ timesteps
7:          Compute advantage function $\hat{A}^{\pi_{\theta + \sigma \epsilon_{i,j}}}$ according to Equation (15)
8:       **end for**
9:       Compute the state-value target $\hat{G}_t$ for each state $s_t$ according to Equation (13)
10:    **end for**
11:    Collect $(s, \hat{G})$ for critic update and $(\epsilon, \hat{A})$ for actor update
12:    Update $w$ with batch size $L$ through SGD by minimizing Equation (14) for $M$ epochs
13:    Update $\theta$ along the zeroth-order gradient direction estimated in Equation (16)
14: **end for**

---

## A.2 IMPLEMENTATION DETAILS

We implemented ZOAC with parallelized workers (Algorithm 1) using the distributed framework Ray (Moritz et al., 2018). We follow the parallelization techniques used in ES (Salimans et al., 2017) and ARS (Mania et al., 2018). Firstly, we created a shared noise table before training starts, then the workers communicate indices in the shared table but not the perturbation vectors, so as to avoid high communication cost. Besides, random seeds for constructing parallelized training environments and the evaluation environment are different and generated from a single seed designated before hand.

We use two different types of policies: linear policies for ARS and ZOAC (linear), neural networks with (64, 64) hidden nodes and tanh nonlinearities for ES, PPO and ZOAC (neural). For actor-critic algorithms, we use neural networks with (256, 256) hidden nodes and tanh nonlinearities as critics to estimate state-value function.

Both the zeroth-order baseline methods perform reward shaping to resolve the local optima problem as described in the original paper: ARS subtracts the survival bonus from rewards (1 in Hopper and Ant), while ES transforms the episodic returns into rankings. ES further discretize the actions to encourage exploration in Hopper but we do not reserve this trick for comparison since discretization will lead to a different policy architecture.

We summarize the hyperparameters used in ZOAC in Table 3 and list their values that are used to produce the results in Figure 2. We tune several important hypermarameters ($n$, $N$, $H$, $\sigma$) via coarse grid search and select the best performing setting to produce the final results. During evaluation, exploration noise are turned off and the reported total average return is averaged over 10 episodes. Table 1 summarizes the maximum value of the total average return within the timestep threshold, averaged over 5 trials.

## A.3 DERIVATION OF THE VARIANCE BOUND

**Theorem 1.** *If the reward* $|r(s, a)| < \alpha$*, the critic network output* $|V_w(s)| < \beta$*, and* $n$ *trajectories with length of* $N \times H$ *timesteps are collected in one iteration, the upper bounds of the variance of gradient estimators (Equation (9) and (10)) are:*

$$\text{Var}(\nabla_{\boldsymbol{\theta}} \hat{J}_{\text{ES}}(\boldsymbol{\theta})) \leq \frac{(1 - \gamma^{NH})^2 \alpha^2 d}{n \sigma^2 (1 - \gamma)^2}$$

$$\text{Var}[\nabla_{\boldsymbol{\theta}} \hat{J}_{\text{ZOAC}}(\boldsymbol{\theta})] \leq \frac{((1 - \gamma^N)\alpha + (1 - \gamma)(1 + \gamma^N)\beta)^2 d}{nH \sigma^2 (1 - \gamma)^2}$$

Table 3: Hyperparameters of ZOAC for the learning curves shown in Figure 2

| Environment | Inv.D.P.-v2 | | Hopper-v2 | | HalfCheetah-v2 | | Ant-v2 | |
|---|---|---|---|---|---|---|---|---|
| Policy type | Linear | Neural | Linear | Neural | Linear | Neural | Linear | Neural |
| Num. of workers $n$ | 4 | 8 | 4 | 8 | 4 | 8 | 8 | |
| Rollout length $N$ | 10 | | 20 | 10 | 20 | | 20 | |
| Train frequency $H$ | 16 | | 16 | 32 | 16 | | 256 | |
| Para. noise std. $\sigma$ | 0.02 | | 0.04 | | 0.04 | 0.06 | 0.02 | |
| Batch size $L$ | 64 | | | | 128 | | | |
| Num. of epochs $M$ | 8 | | | | 4 | | | |
| Actor optimizer | Adam($\alpha_{\text{actor}} = 0.005, \beta_1 = 0.9, \beta_2 = 0.999$) | | | | | | | |
| Critic optimizer | Adam($\alpha_{\text{critic}} = 0.0003, \beta_1 = 0.9, \beta_2 = 0.999$) | | | | | | | |
| Discount factor $\gamma$ | 0.99 | | | | | | | |
| GAE coeff. $\lambda$ | 0.95 | | | | | | | |

*Proof.* (1) Variance bound for ES gradient estimators

Under the setting described in Section 3.2, the state-value under policy $\pi_{\boldsymbol{\theta}+\sigma\boldsymbol{\epsilon}}$ is estimated by the accumulative return over $NH$ timesteps, which is denoted as $\hat{V}_{NH}^{\pi_{\boldsymbol{\theta}+\sigma\boldsymbol{\epsilon}}}$. The isotropic Gaussian noise added to the policy can be presented as $\boldsymbol{\epsilon} = (\epsilon_1, \epsilon_2, ..., \epsilon_d)^\top$, where $\epsilon_l \sim \mathcal{N}(0,1), l \in \{1, 2, ..., d\}$.

$$
\begin{aligned}
\text{Var}[\hat{V}_{NH}^{\pi_{\boldsymbol{\theta}+\sigma\boldsymbol{\epsilon}}}\boldsymbol{\epsilon}] &\leq \sum_{l=1}^{d} \mathbb{E}[(\hat{V}_{NH}^{\pi_{\boldsymbol{\theta}+\sigma\boldsymbol{\epsilon}}}\epsilon_l)^2] \\
&= \sum_{l=1}^{d} \int p(\epsilon_l) \left(\sum_{t=1}^{NH} \gamma^{t-1} r(s_t, a_t)\right)^2 \epsilon_l^2 \mathrm{d}\epsilon_l \\
&\leq \sum_{l=1}^{d} \int p(\epsilon_l) \left(\sum_{t=1}^{NH} \gamma^{t-1} \alpha\right)^2 \epsilon_l^2 \mathrm{d}\epsilon_l \\
&= \frac{(1-\gamma^{NH})^2 \alpha^2}{(1-\gamma)^2} \sum_{l=1}^{d} \int p(\epsilon_l) \epsilon_l^2 \mathrm{d}\epsilon_l \\
&= \frac{(1-\gamma^{NH})^2 \alpha^2}{(1-\gamma)^2} \sum_{l=1}^{d} \mathbb{E}_{\epsilon_l \sim \mathcal{N}(0,1)} \epsilon_l^2 \\
&= \frac{(1-\gamma^{NH})^2 \alpha^2 d}{(1-\gamma)^2}
\end{aligned}
$$

The last equality holds because $\epsilon_l^2 \sim \chi^2(1)$ when $\epsilon_l \sim \mathcal{N}(0,1)$, and $\mathbb{E}[\epsilon_l^2] = 1$ for all $l$. Since $n$ random directions is sampled and evaluated, the ES gradient estimator is given according to Equation (9):

$$
\nabla_{\boldsymbol{\theta}} \hat{J}_{\text{ES}}(\boldsymbol{\theta}) = \frac{1}{n\sigma} \sum_{i=1}^{n} \hat{V}_{NH}^{\pi_{\boldsymbol{\theta}+\sigma\boldsymbol{\epsilon}_i}} \boldsymbol{\epsilon}_i
$$

Therefore the variance bound for ES can be derived as in Theorem 1:

$$
\begin{aligned}
\text{Var}[\nabla_{\boldsymbol{\theta}} \hat{J}_{\text{ES}}(\boldsymbol{\theta})] &= \frac{1}{n\sigma^2} \text{Var}[\hat{V}_{NH}^{\pi_{\boldsymbol{\theta}+\sigma\boldsymbol{\epsilon}}}\boldsymbol{\epsilon}] \\
&\leq \frac{(1-\gamma^{NH})^2 \alpha^2 d}{n\sigma^2 (1-\gamma)^2}
\end{aligned}
$$

(2) Variance bound for ZOAC gradient estimators

Under the setting described in Section 3.2, the performance under policy $\pi_{\boldsymbol{\theta}+\sigma\boldsymbol{\epsilon}}$ is estimated by the $N$-step TD residual, which is denoted as $\hat{A}_N^{\pi_{\boldsymbol{\theta}+\sigma\boldsymbol{\epsilon}}}$. The isotropic Gaussian noise $\boldsymbol{\epsilon}$ is added to the policy as well.

$$
\begin{aligned}
\mathrm{Var}[\hat{A}_N^{\pi_{\boldsymbol{\theta}}+\sigma\boldsymbol{\epsilon}}\boldsymbol{\epsilon}] &\leq \sum_{l=1}^{d} \mathbb{E}[(\hat{A}_N^{\pi_{\boldsymbol{\theta}}+\sigma\boldsymbol{\epsilon}}\epsilon_l)^2] \\
&= \sum_{l=1}^{d} \int p(\epsilon_l) \left( \sum_{t=1}^{N} \gamma^{t-1} r(s_t, a_t) + \gamma^N V_w(s_{t+N}) - V_w(s_t) \right)^2 \epsilon_l^2 \mathrm{d}\epsilon_l \\
&\leq \sum_{l=1}^{d} \int p(\epsilon_l) \left( \sum_{t=1}^{N} \gamma^{t-1}\alpha + (1+\gamma^N)\beta \right)^2 \epsilon_l^2 \mathrm{d}\epsilon_l \\
&= \left( \frac{(1-\gamma^N)\alpha + (1-\gamma)(1+\gamma^N)\beta}{(1-\gamma)} \right)^2 \sum_{l=1}^{d} \int p(\epsilon_l)\epsilon_l^2 \mathrm{d}\epsilon_l \\
&= \left( \frac{(1-\gamma^N)\alpha + (1-\gamma)(1+\gamma^N)\beta}{(1-\gamma)} \right)^2 d
\end{aligned}
$$

Totally $n \times H$ random directions is sampled and evaluated, and the ZOAC gradient estimator is given according to Equation (10):

$$
\nabla_{\boldsymbol{\theta}} \hat{J}_{\mathrm{ZOAC}}(\boldsymbol{\theta}) \approx \frac{1}{nH\sigma} \sum_{i=1}^{nH} \hat{A}_N^{\pi_{\boldsymbol{\theta}}+\sigma\boldsymbol{\epsilon}_i} \boldsymbol{\epsilon}_i
$$

Therefore the variance bound for ZOAC can be derived:

$$
\begin{aligned}
\mathrm{Var}[\nabla_{\boldsymbol{\theta}} \hat{J}_{\mathrm{ZOAC}}(\boldsymbol{\theta})] &= \frac{1}{nH\sigma^2} \mathrm{Var}[\hat{A}_N^{\pi_{\boldsymbol{\theta}}+\sigma\boldsymbol{\epsilon}}\boldsymbol{\epsilon}] \\
&\leq \frac{((1-\gamma^N)\alpha + (1-\gamma)(1+\gamma^N)\beta)^2 d}{nH\sigma^2(1-\gamma)^2}
\end{aligned}
$$

$\square$

### A.4 STATE-VALUE FUNCTION ESTIMATION

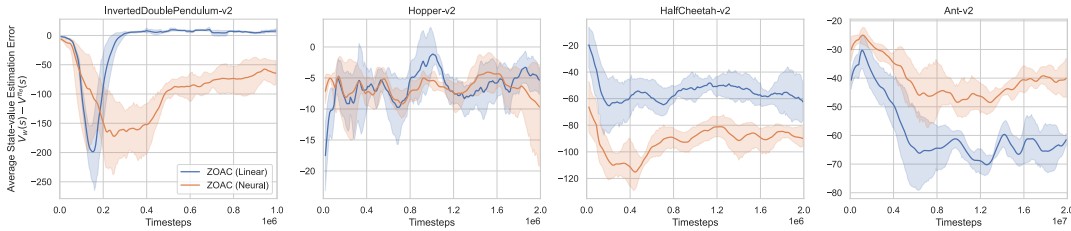

Figure 6: Average state-value estimation difference $V_w(s) - V^{\pi_\theta}(s)$ in evaluation during training in Figure 2. The solid lines correspond to the mean and the shaded regions to the 95% confidence interval over 5 trials using a fixed set of random seeds. All curves are smoothed uniformly for visual clarity.

We plot the average state-value estimation difference $V_w(s) - V^{\pi_\theta}(s)$ in evaluation during training in Figure 6. Since we turn off the exploration noise for evaluation, which means that the trajectories are collected under the deterministic policy $\pi_\theta$, the discounted sum of reward-to-go can be regarded as an estimate of the true state-value.

Results show that the critic networks converge, but in most cases to an underestimated value. This is because the critic network is trained to fit the state-value function $V^\beta(s)$ of the stochastic exploration policy $\beta$ rather than $V^{\pi_\theta}(s)$ of the deterministic policy $\pi_\theta$. The underestimate bias vary in different tasks and when using different forms of policies, which is related to the local shape of the optima found by the RL agent. However, due to the objective function used in ZOAC, intuitively, the agent tend to find wide optima during training, which finally result in more robust policies.

A.5 Additional results of robustness comparison

Table 4 shows the result of robustness comparison in all tested MuJoCo environments. Results show that in general neural policies are more robust, which can be ascribed to their strong expressive ability. InvertedDoublePendulum-v2 is the exception, in which the robustness of the neural network policies is no where near the linear ones, no matter what RL algorithms is used. Since this environment is the simplest one among these environments, we guess that this is due to the overfitting of neural networks, and that a matrix is enough to represent an optimal policy.

When comparing between linear policies, the policies learned by ZOAC yield higher robustness to both observation noise and parameter noise in all environments, compared to those learned by ARS. One possible reason is that ARS subtracts the survival bonus from rewards, which actually alter the objective function being optimized, as described in Section 4.1. As for neural policies, the policies learned by ZOAC are also shown to be more robust to both observation noise and parameter noise. One reason is that parameter noise used in zeroth-order methods encourages the agent to find a wide optima which is robust to parameter perturbations, while gradient-based methods focus on the performance of a particular point. Besides, ZOAC perform timestep-wise perturbation rather than episode-wise perturbation, which explores along more random directions and collects more diverse trajectories (i.e., data) than ARS or ES. All these features finally lead to better generalization and robustness of the policies learned by ZOAC.

Table 4: Robustness comparison of the learned policies. Best performing policies from Figure 2 are tested (4e7 timestep limitation for ARS and ES in HalfCheetah-v2 and Ant-v2 till convergence). Each policy is evaluated on 400 trajectories in the same environment (using the same random seed that has never been used during training) and the average return over 5 policies are listed.

| Env. | Noise type | No extra noise | Obs. noise ($\sigma = 0.1$) | Para. noise ($\sigma = 0.05$) |
|---|---|---|---|---|
| Inv.D.P. | **ZOAC(Linear)** | 8744.11 | **8615.50 (-1.5%)** | **3643.49 (-58.3%)** |
| | **ZOAC(Neural)** | 8419.16 | 908.95 (-89.2%) | 1344.54 (-84.0%) |
| | ARS(Linear) | 8883.13 | **8339.77 (-6.1%)** | 3396.74 (-61.8%) |
| | ES(Neural) | 7333.34 | 3593.76 (-51.0%) | 927.02 (-87.4%) |
| | PPO(Neural) | 7477.48 | 3537.34 (-52.7%) | 3115.82 (-58.3%) |
| Hopper | **ZOAC(Linear)** | 2885.10 | **2733.24 (-5.3%)** | **1872.07 (-35.1%)** |
| | **ZOAC(Neural)** | 2417.33 | **2284.12 (-5.5%)** | **1644.34 (-32.0%)** |
| | ARS(Linear) | 2587.06 | 1194.43 (-53.8%) | 781.99 (-69.8%) |
| | ES(Neural) | 1443.77 | 1330.98 (-7.8%) | 915.38 (-36.6%) |
| | PPO(Neural) | 2596.02 | 2185.02 (-15.8%) | 395.80 (-84.8%) |
| HalfC. | **ZOAC(Linear)** | 4909.53 | 4402.54 (-10.3%) | 3289.15 (-33.0%) |
| | **ZOAC(Neural)** | 5179.39 | **5057.07 (-2.4%)** | **4247.08 (-18.0%)** |
| | ARS(Linear) | 4529.30 | 1771.23 (-60.9%) | 1724.40 (-61.9%) |
| | ES(Neural) | 5478.01 | 4550.22 (-16.9%) | 1527.93 (-72.1%) |
| | PPO(Neural) | 4668.55 | 3342.23 (-28.4%) | 972.08 (-79.2%) |
| Ant | **ZOAC(Linear)** | 4134.79 | 3460.67 (-16.3%) | -2901.51 (-170.2%) |
| | **ZOAC(Neural)** | 4013.57 | **3650.79 (-9.0%)** | **976.57 (-75.6%)** |
| | ARS(Linear) | 3749.73 | 2868.18 (-23.5%) | -4269.38 (-213.9%) |
| | ES(Neural) | 4029.82 | **3902.70 (-3.2%)** | **970.30 (-75.9%)** |
| | PPO(Neural) | 3103.79 | 2792.79 (-10.0%) | 393.20 (-87.3%) |

A.6 Performance comparison of different policy parameterizations

The derivative-free nature of ZOAC allows us to estimate the zeroth-order policy gradient to improve the policy without considering the specific policy architecture. Hence, ZOAC can be applied seamlessly to arbitrary parameterized policies in theory.

We further apply ZOAC on two more different policies and conduct additional experiments on the same environments with four different policies, from fewer parameters to more parameters: Toeplitz matrix, matrix, network with (64, 64) units, network with (128, 128) units, and to see the performance of ZOAC when optimizing these policies. We listed the dimension of the parameter space of different policies in Table 5.

Toeplitz matrix is a kind of compact policies with parameter sharing schemes, each element depends only on the difference between the row index and the column index. A general dense matrix $M \in \mathbb{R}^{m \times n}$ has $m \times n$ parameters, while a Toeplitz matrix $T \in \mathbb{R}^{m \times n}$ has only $m + n - 1$ parameters. Since a general dense matrix performs quite well in all tested environments, we wonder whether a more compact policy still work. Besides, as derived in Theorem 1, the variance bound of zeroth-order gradient increases proportional to the dimension of parameter space, and it is a common view that zeroth-order methods are more suitable for low-dimensional problems. We wonder whether a larger network will improve or harm the performance.

We use the same set of hyperparameters for linear policies, and another set for networks, as summarized in Table 3. Results in Figure 7 show that a Toeplitz matrix can obtain average return around 2000 in challenging environments like HalfCheetah-v2 and Ant-v2 with only very few parameters. However, it is far inferior to a general dense matrix, indicating that this type of policy is not sufficient to represent an optimal policy. Network with (64, 64) and (128, 128) hidden nodes performs quite similarly to each other.

The additional results demonstrate the wide adaptability of ZOAC to different forms of policies. Zeroth-order policy update makes it very useful when gradient information is hard to obtain or even unavailable, like low precision neural networks, hierarchical policies, or even rule-based controllers. In the future, we may apply ZOAC to specific problems where first-order methods can not handle and to improve existing results where the policies are trained with traditional zeroth-order methods like ES and ARS.

Table 5: Dimension of the state space, action space and parameter space of different policies.

| Environment | Inv.Dou.Pen.-v2 | Hopper-v2 | HalfCheetah-v2 | Ant-v2 |
|---|---|---|---|---|
| State space | 11 | 11 | 17 | 111 |
| Action space | 1 | 3 | 6 | 8 |
| Toeplitz matrix | 11 | 13 | 22 | 118 |
| Matrix | 11 | 33 | 102 | 888 |
| Network (64, 64) | 4993 | 5123 | 5702 | 11848 |
| Network (128, 128) | 18177 | 18435 | 19590 | 31880 |

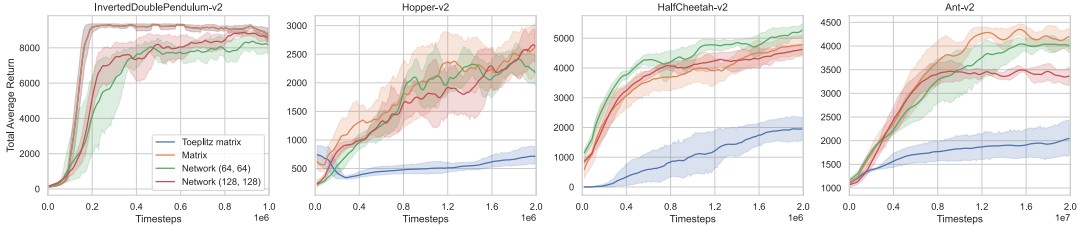

Figure 7: Learning curves on MuJoCo benchmarks with different policies. The solid lines correspond to the mean and the shaded regions to the 95% confidence interval over 5 trials using a fixed set of random seeds. All curves are smoothed uniformly for visual clarity.

