# OpenReview forum: "Zeroth-Order Actor-Critic"
_ICLR.cc/2022/Conference — ICLR 2022 Submitted_

### Official Review · Reviewer_wk6f · 2021-10-20

**Correctness:** 3
**Technical Novelty And Significance:** 3
**Empirical Novelty And Significance:** 3
**Recommendation:** 6
**Confidence:** 3

**Main Review:**

The paper is overall well-written, and both the actor and critic part of the algorithms are relatively easy to understand (especially with the pseudocode provided in appendix). The algorithm makes sense, and the results are convincing. The introduction does a great job at motivating why a zero-order policy update is desirable, even though the list provided in paragraph 2 could be made more prominent.

One small limitation of the paper is that two extra baselines would have been interesting in the experiments: SAC and A3C. PPO is quite a dated baseline, and does not represent state-of-the-art first-order reinforcement learning. The Soft Actor-Critic has been shown to outperform PPO, and other old baselines, such as A3C, may be interesting too as they use several workers to collect rollouts (leading to better exploration).

A minor remark is that Figure 1 did not help me understand the algorithm, while the pseudocode helped better. I suggest to mention how $\hat{G}$ and $\hat{A}$ are computed in Figure 1a (and remove the loop with the environment to save space), noting that they come from rollouts computed in parallel with each rollout having its own perturbed policy parameters. Then, I would replace the small lines like $V_{\omega} \rightarrow V_{\omega'}$ with the actual gradients being followed (maybe in a simplified way), so that we clearly see that $\hat{G}$ and $\hat{A}$ are used to train the critic and the actor, and that training the actor does not require a gradient computation anymore, but is a weighted sum of perturbed parameters. It is challenging to draw this well, I agree, but I think that a clear drawing will be very useful if this paper is accepted and has to be presented at the conference.

**Summary Of The Paper:**

The paper proposes an actor-critic reinforcement learning algorithm, compatible with continuous actions, that combines a Monte-Carlo computation of the gradient of the actor (based on workers that perform rollouts from the current state) with a standard supervised-learning based critic update rule (using gradient descent for several epochs). The core contribution of the paper seems to be the method that combines the critic network V(s) with on-policy samples, using a form of eligibility traces. This combination is used to compute target V-Values for the critic, and the weighted average used to compute the gradient of the actor. Empirical results on MuJoCo show that the proposed algorithm outperforms PPO in sample-efficiency and final policy quality (extra experiments show that robustness is high too).

**Summary Of The Review:**

The paper presents a well-motivated and elegant idea to train an actor-critic algorithm, and the empirical evaluation is sufficient. I'm borderling recommending acceptance (I would be happy to see the paper accepted, but could live with it being rejected in case I missed anything). I think that more recent baselines would make the paper stronger: strong theory and motivation, then a strong comparison against state-of-the-art baselines.

---

> ### Author Response · Authors · 2021-11-15
> **Author Response to Reviewer wk6f**
>
> We sincerely thank you for thorough reviews and insightful comments.
>
> > One small limitation of the paper is that two extra baselines would have been interesting in the experiments: SAC and A3C. PPO is quite a dated baseline, and does not represent state-of-the-art first-order reinforcement learning. The Soft Actor-Critic has been shown to outperform PPO, and other old baselines, such as A3C, may be interesting too as they use several workers to collect rollouts (leading to better exploration).
>
> ZOAC aims to unify zeroth-order and first-order methods into an on-policy actor-critic architecture and to preserve the advantages from both. Therefore, we mainly focus on comparison between on-policy RL algorithms due to their similarities.  As far as we know, PPO is one of the best performing and widely-used on-policy RL algorithms and therefore is chosen as the first-order baseline. As for A3C, as illustrated in [1], the synchronous variant A2C was found to have the same or better performance than the asynchronous version A3C. In [1], PPO was shown to outperforms A2C on all seven tested MuJoCo environments. Therefore, we think that comparing with PPO is sufficient to demonstrate the advantage of ZOAC over first-order on-policy RL algorithms.
>
> As for off-policy actor-critic algorithms like SAC which stem from Q-learning, they are a little bit different from on-policy actor-critic algorithms like PPO that stem from policy gradient. We agree that a fine-tuned version of SOTA off-policy algorithms may achieve high sample efficiency with the usage of replay buffer, and we indeed consider to combine experience replay into zeroth-order policy improvement so as to further improve sample efficiency while still maintaining the specific advantages of ZOAC which have been proven in this paper. However, we leave it as our future work and at present we mainly focus on integrating zeroth-order policy update into an on-policy actor-critic architecture, and therefore we only choose the most related methods as baselines in our paper.
>
> We list some results from previous paper below as a reference. One of the chosen zeroth-order baseline, ARS, compared with DDPG and SAC in their original paper. In Table 4 & Table 5 in Appendix A.2 in [2], ARS outperforms DDPG in 3 out of 4 MuJoCo environments, SAC in 2 out of 4, and A2C in 3 out of 4. In our experiments, ARS performs consistently with the results listed in their own paper, and ZOAC have the same or better performance than ARS both in sample efficiency, final performance, variance over trials, and also robustness of the learned policies.
>
> > A minor remark is that Figure 1 did not help me understand the algorithm, while the pseudocode helped better.
>
> Thank you so much for your revision suggestions on the schematic diagram of ZOAC. We have modified Figure 1a to make it more clear and informative based on your suggestions.
>
> [1] Schulman, John, et al. "Proximal policy optimization algorithms." *arXiv preprint arXiv:1707.06347* (2017).
>
> [2] Mania, Horia, Aurelia Guy, and Benjamin Recht. "Simple random search of static linear policies is competitive for reinforcement learning." *Proceedings of the 32nd International Conference on Neural Information Processing Systems*. 2018.
>
> We hope that our response has addressed your concerns. If you still have other comments on our work, or if we have misunderstood your concerns, please let us know and we would be happy to make further response or revision.

---

> > ### Comment · Reviewer_wk6f · 2021-11-21
> > **Nice updated figure**
> >
> > Thank you for the information and the update. The figure is indeed more informative now (still not enough to understand the algorithm, but it is fully normal, as this is why we write papers and don't present algorithms just with figures). We now see all the components and where they intervene.
> >
> > I also understand your explanation of why you did not compare against SAC. I see the logic here, and the related results help position the paper. My main suggestion is that very interested and enthusiastic readers of the paper may want to know that the proposed algorithm not only outperforms state-of-the-art on-policy algorithms, but also off-policy ones.

---

### Official Review · Reviewer_eX28 · 2021-10-25

**Correctness:** 3
**Technical Novelty And Significance:** 3
**Empirical Novelty And Significance:** 3
**Recommendation:** 6
**Confidence:** 4

**Main Review:**

This paper developed a very interesting zeroth-order actor-critic algorithm that nicely integrates gradient-based critic training with gradient-free actor training. The newly developed algorithm was also evaluated empirically on several benchmark datasets with promising outcomes.

While this paper introduced some interesting ideas, there are some limitations too:

1. It remains questionable to me why the new algorithm is designed to integrate first-order PEV with zeroth-order PIM. In fact, it seems mathematically possible to induce first-order PIM based on first-order PEV. I am not convinced about the technical and practical necessity of adopting zeroth-order PIM instead of first-order PIM. Perhaps the performance advantage of zero-order PIM over first-order PIM should be experimentally and theoretically evaluated in further details.

2. The new algorithm introduces additional hyper-parameters, including N. While the performance impact of N has been evaluated on one benchmark problem in detail, it is not clear how to set this hyper-parameter properly for other reinforcement learning problems. Some further investigation on N may be necessary.

3. The theoretical analysis in Subsection 3.2 produced two upper bounds on the variances of the zeroth-order gradient. Although the upper bound for the new zeroth-order gradient may be smaller than that of the existing zeroth-order gradient based on ES, this does not necessarily imply that the new zeroth-order gradient actually enjoys smaller variance than ES during practical use. In consideration of the bias introduced by the critic, the true advantage of the new algorithm may require further investigation.

4. The experimental evaluation may need to be strengthened in several aspects. First, the authors should consider more challenging benchmarks including Humanoid in order to clearly show the performance advantage of the new algorithm. Second, the new algorithm is only compared to one first-order approach, i.e., PPO. This is not sufficient to demonstrate the effectiveness of the new algorithm. Comparing to other first-order approaches, such as SAC and TD3, is necessary IMHO. Third, based on the learning curves and the results presented in table 1, the performance of ZOAC is not significantly better than PPO. With suitable parameter tuning, PPO may easily outperform ZOAC on the tested benchmark problems. Hence, the true effectiveness of ZOAC should be studied further.

Additional comments based on authors' feedback:
I would like to thank the authors for providing feedback on my review comments. Following the feedback provided, I see the necessity of mentioning the following:

a. In the feedback, the authors highlighted several advantages of ZOO, including its capability to train policy where gradient information is hard to obtain or even unavailable. This is a good point. However, this paper does not show that the newly proposed algorithm can effectively train such a policy, for example a rule-based policy, to solve any practically important problems. Furthermore, the robustness advantage of ZOO, in comparison to first-order PIM, should be more clearly analyzed in the paper.

b. My concern is that the bounds identified in the paper may not be sufficiently tight. Hence, comparing the bounds directly does not show the true theoretical advantage of the new algorithm, affecting the theoretical contribution of this paper.

c. Thank the authors for agreeing to perform more experiments, including experiments on Humanoid. However, without knowing the experiment results, it is not easy to determine the true advantage of the new algorithm on hard problems at this stage.

d. It is widely known that the performance of deep reinforcement learning algorithms can be highly sensitive to detailed experimental settings. While ARS has been shown to outperform DDPG and SAC previously, this may not be sufficient to avoid comparing the new algorithm with DDPG, SAC or TD3. As carefully pointed out by the authors, the performance results reported in the paper introducing TD3 may not be identical to those presented in the paper introducing SAC. Hence, to truly understand the performance advantage of the new algorithm and hence the technical contribution of this paper, I believe the authors should consider comparing their algorithm with more SOTA algorithms, including more gradient-based algorithms.

**Summary Of The Paper:**

This paper developed a very interesting zeroth-order actor-critic algorithm that nicely integrates gradient-based critic training with gradient-free actor training. The newly developed algorithm was also evaluated empirically on several benchmark datasets with promising outcomes. The literature review in the paper is very clear and comprehensive. The review adequately justified the technical innovation of the new algorithm.

**Summary Of The Review:**

This paper developed a very interesting zeroth-order actor-critic algorithm that nicely integrates gradient-based critic training with gradient-free actor training. While this paper introduced some interesting ideas, the motivation for the new algorithm design may need to be clarified more. The theoretical strength of the new algorithm should be analyzed in further depth. There are rooms for improvement regarding the experimental evaluation too.

---

> ### Author Response · Authors · 2021-11-15
> **Author Response to Reviewer eX28**
>
> We are really appreciated for the detailed and insightful comments. Below are our response.
>
> > I am not convinced about the technical and practical necessity of adopting zeroth-order PIM instead of first-order PIM.  Perhaps the performance advantage of zero-order PIM over first-order PIM should be experimentally and theoretically evaluated in further details.
>
> Zeroth-order update of course does not outperform first-order update in all aspects but it has its own advantages. We describe several attractive advantages of zeroth-order optimization when applied to RL problems in the introduction section. Firstly, ZOO is not restricted to differentiable policies, and would be very useful when gradient information is hard to obtain or even unavailable, like low precision neural network, hierarchical policy, or even rule-based controllers. The derivative-free nature allows us to estimate the zeroth-order gradient of return w.r.t. policy parameters without using the specific policy architecture. ZOO have been used successfully in these kind of RL problems, like [1] [2] [3]. Secondly, exploration in parameter space is more suitable then action space exploration for long horizon control problems with low dimensional policy parameterization, as demonstrated in [4]. Thridly, Zeroth-order population based optimization possesses robustness-seeking property and diverse policy behaviors, as demonstrated in [5].  The main limitation of existing methods is sample efficiency and our main contribution is that we unify these two methods into an on-policy actor-critic architecture to increase sample efficiency while still preserving the above-mentioned advantages of zeroth-order PIM. In the future, we may apply ZOAC to those cases where first-order methods can not handle and to improve existing results where the policies are trained with traditional zeroth-order methods.
>
> > The new algorithm introduces additional hyper-parameters, including N. While the performance impact of N has been evaluated on one benchmark problem in detail, it is not clear how to set this hyper-parameter properly for other reinforcement learning problems.
>
> We derive the variance bound of ZOAC in Theorem 1 and then give an intuitive comparison. A smaller N in ZOAC leads to smaller variance, but should not be too small, otherwise the bias of critic may harm the performance. Empirically, during hyperparameter tuning, we found that the rollout length $N$ is actually not very sensitive. A large range of $N$, approximately from 5 to 50, perform quite well across tasks. We list the final hyperparameter settings in Table 3 in Appendix A.2, in which the chosen $N$ is around 10 to 20 in the tested MuJoCo environments.
>
> > Although the upper bound for the new zeroth-order gradient may be smaller than that of the existing zeroth-order gradient based on ES, this does not necessarily imply that the new zeroth-order gradient actually enjoys smaller variance than ES during practical use. In consideration of the bias introduced by the critic, the true advantage of the new algorithm may require further investigation.
>
> We derive the variance bound theoretically, and give a intuitive comparison in Section 3.2. In experiments, ZOAC outperforms ES across tasks in learning speed, final performance, and variance over trials, including high-dimensional challenging tasks like Ant. We think that present result is sufficient to demonstrate the advantage of ZOAC over ES. However, we agree with you that a more thorough and rigorous comparison will be helpful and we leave it as our future work.

---

> > ### Author Response · Authors · 2021-11-15
> > **Author Response to Reviewer eX28 (cont'd)**
> >
> >
> > > The experimental evaluation may need to be strengthened in several aspects:
> > >
> > > (1) More challenging benchmarks including Humanoid;
> > >
> > > (2)  SAC and TD3 can be added as baselines;
> > >
> > > (3) The advantage over PPO is not significant, with suitable parameter tuning, PPO may outperform ZOAC;
> >
> > (1) Thank you for your advice. We will run additional experiments on Humanoid and the results will be added to the paper when finished.
> >
> > (2) Off-policy actor-critic algorithms like SAC which stem from Q-learning, they are a little bit different from on-policy actor-critic algorithms like PPO that stem from policy gradient. We agree that a fine-tuned version of SOTA off-policy algorithms may achieve high sample efficiency with the usage of replay buffer, and we indeed consider to combine experience replay into zeroth-order policy improvement so as to further improve sample efficiency while still maintaining the specific advantages of ZOAC which have been proven in this paper. However, we leave it as our future work and at present we mainly focus on integrating zeroth-order policy update into an on-policy actor-critic architecture, and therefore we only choose the most related methods as baselines in our paper.
> >
> > We list some results from previous paper below as a reference. One of the chosen zeroth-order baseline, ARS, compared with DDPG and SAC in their original paper. In Table 4 & Table 5 in Appendix A.2 in [6], ARS outperforms DDPG in 3 out of 4 MuJoCo environments, SAC in 2 out of 4. In our experiments, ARS performs consistently with the results listed in their own paper, and ZOAC have the same or better performance than ARS both in sample efficiency, final performance, variance over trials, and also robustness of the learned policies. Besides, one thing interesting is that, as two concurrent works, TD3 and SAC both reported very different results in their original paper: TD3 have the same or better performance than SAC in all 7 MuJoCo environments in TD3's paper (Figure 5 and Table 1 in [7]), while SAC outperforms TD3 in 4 out of 5 in SAC's paper (Figure 1 in [8]).
> >
> > (3) We have chosen a stable and efficient implementation of PPO and follow the recommended hyperparameter settings, and actually the performance is better than the results shown in the original paper [9] and also the results shown in the ARS paper [6] (one of our baseline). We believe this is already a representative result for PPO. In addition to the sample efficiency and final performance, we have already shown other advantages of ZOAC over PPO in the paper, including robustness and exploration, which stem from the derivative-free nature of the zeroth-order policy improvement and the state-dependent and temporally extended exploration with parameter noise.
> >
> > [1] Jain, Deepali, Atil Iscen, and Ken Caluwaerts. "Hierarchical reinforcement learning for quadruped locomotion." *2019 IEEE/RSJ International Conference on Intelligent Robots and Systems (IROS)*. IEEE, 2019.
> >
> > [2] Jain, Deepali, Atil Iscen, and Ken Caluwaerts. "From pixels to legs: Hierarchical learning of quadruped locomotion." *arXiv preprint arXiv:2011.11722* (2020).
> >
> > [3] Likmeta, Amarildo, et al. "Combining reinforcement learning with rule-based controllers for transparent and general decision-making in autonomous driving." *Robotics and Autonomous Systems* 131 (2020): 103568.
> >
> > [4] Vemula, Anirudh, Wen Sun, and J. Bagnell. "Contrasting exploration in parameter and action space: A zeroth-order optimization perspective." *The 22nd International Conference on Artificial Intelligence and Statistics*. PMLR, 2019.
> >
> > [5] Lehman, Joel, et al. "ES is more than just a traditional finite-difference approximator." *Proceedings of the Genetic and Evolutionary Computation Conference*. 2018.
> >
> > [6] Mania, Horia, Aurelia Guy, and Benjamin Recht. "Simple random search of static linear policies is competitive for reinforcement learning." *Proceedings of the 32nd International Conference on Neural Information Processing Systems*. 2018.
> >
> > [7] Fujimoto, Scott, Herke Hoof, and David Meger. "Addressing function approximation error in actor-critic methods." *International Conference on Machine Learning*. PMLR, 2018.
> >
> > [8] Haarnoja, Tuomas, et al. "Soft actor-critic algorithms and applications." *arXiv preprint arXiv:1812.05905* (2018).
> >
> > [9] Schulman, John, et al. "Proximal policy optimization algorithms." *arXiv preprint arXiv:1707.06347* (2017).
> >
> > We hope that our response has addressed your concerns. If you still have other comments on our work, or if we have misunderstood your concerns, please let us know and we would be happy to make further response or revision.

---

> > > ### Comment · Reviewer_cTPK · 2021-11-15
> > > **Re: Author Response to Reviewer eX28**
> > >
> > > It seems I misunderstood the paper, and reading the authors' response, I am increasing the score accordingly.

---

### Official Review · Reviewer_cTPK · 2021-10-31

**Correctness:** 1
**Technical Novelty And Significance:** 2
**Empirical Novelty And Significance:** 2
**Recommendation:** 6
**Confidence:** 3

**Main Review:**

While it is a novel choice to sample a trajectory consisting of a number of perturbed different policies, I am not sure whether we can compute a valid value function out of these trajectories. in Eq. 14, we are learning a single critic over return estimates from many different perturbed policy samples. What policy is this value function computes value for? For accurate advantage computation and policy update in Eq. 15/16, the advantage should be computed with the value function for each perturbed, policy; however, It seems that the paper is learning a single critic over all perturbed policies and computing the zeroth-order gradient based on it. This may be OK for Mujoco-like domains where the optimizing the short-term sum of rewards is enough to get an optimal policy, but it can be problematic in the domains where long-term reward is important. It is a very important point on the correctness of the main algorithm, and I may change my position if the authors can address this concern.

On the empirical evaluations, it is interesting to have experiments showing various strengths of ZOAC, especially its robustness and ablations. However, the overall performance evaluations do not seem very competitive when compared to sota actor-critic algorithms, e.g. SAC can achieve a return over 10000 at 1e6, and it would be much better to include such algorithms for comparison. Actor with (64,64) hidden units is also not a standard choice, and it would be more interesting to see the results with different number of units, e.g. (128,128) and (256,256).

**Summary Of The Paper:**

This paper proposes ZOAC, which is composed of: perturbed rollouts generation, first-order policy evaluation, and zeroth-order policy improvement. The paper proposes a new way of sampling trajectories, by using a policy only for a fixed number of steps and changing to another policy. The paper shows that we can reduce the variance of the zeroth-order gradient estimator by appropriately setting the number of steps to use the same policy. The empirical results show that ZOAC can outperform other zeroth-order and first-order baseline algorithms.

**Summary Of The Review:**

For now, I have an important concern on the main algorithm and I will recommend rejection unless the authors address my concern.


--------------------------

I have read the authors' response, and I have increased the score accodingly.

---

> ### Author Response · Authors · 2021-11-15
> **Author Response to Reviewer cTPK**
>
> We sincerely thank you for the comprehensive comments on our work and we give our response below.
>
> > I am not sure whether we can compute a valid value function out of these trajectories. in Eq. 14, we are learning a single critic over return estimates from many different perturbed policy samples. What policy is this value function computes value for?
>
> The single critic $V_w(s)$ estimates the state-value $V^\beta$ of the stochastic behavior policy $\beta=\pi_{\theta+\sigma\epsilon}$, which contains Gaussian parameter noise $\epsilon\sim\mathcal{N}(0,I)$ for exploration. The rollout data used for training is collected by the behavior policy $\beta$, which satisfies the on-policy distribution of $\beta$. The value target $\hat{G}$ for each state $s$ is calculated based on the trajectories till the end, rather than within $N$-step segment. In first-order PEV, the single critic network is trained to estimate the state value $V^\beta$ in a supervised learning way, with the on-policy data $(s,\hat{G})$.
>
> > For accurate advantage computation and policy update in Eq. 15/16, the advantage should be computed with the value function for each perturbed, policy; however, It seems that the paper is learning a single critic over all perturbed policies and computing the zeroth-order gradient based on it. This may be OK for Mujoco-like domains where the optimizing the short-term sum of rewards is enough to get an optimal policy, but it can be problematic in the domains where long-term reward is important.
>
> The objective function of ZOAC is illustrated in Eq. (6) that we aim to maximize the expected state-value of the behavior policy $\beta$, which contains parameter noise all the time. We believe that this objective function encourage sufficient state-dependent exploration, and also tend to increase the robustness to parameter noise of the learned policies. Experimental results support our views.
>
> Under this objective, we can derive the zeroth-order policy gradient as in Eq. (8), in which the advantage of each perturbation is calculated as the $N$-step TD residual based on the state-value of the stochastic behavior policy $V^\beta$ rather than the state-value of each corresponding perturbed policy (which is deterministic). That's why we can use the single critic network output $V_w$ (which is trained to approximate $V^\beta$) in the zeroth-order policy improvement.
>
> In the practical algorithm, we follow GAE to introduce $\lambda$ to further control the bias-variance trade-off between multi-step TD target (from 1-step to $N$-step). Besides, in the ablation study we show that the critic network is a crucial part of ZOAC, i.e., N-step accumulative reward alone (without bootstrapping) is not sufficient to guide policy improvement.
>
> In Appendix A.4, we plot the average state-value estimation error $V_w(s)-V^{\pi_\theta}(s)$, which is estimated in evaluation during training. Since we turn off the exploration noise for evaluation, which means that the trajectories are collected under the deterministic policy $\pi_\theta$, the discounted sum of reward-to-go can be regarded as an estimate of the true state-value. The error usually do not converge to zero because the critic is trained to fit the state-value function of the stochastic behavior policy with parameter noise. The underestimate bias vary in different tasks and when using different forms of policies, which is related to the flatness of the optima found by the agent. However, due to the objective function used in ZOAC, intuitively, the agent tend to find wide optima during training, which finally result in more robust policies.
>
> Therefore, the correctness of our algorithm is guaranteed theoretically, which doesn't change with the time horizon of the reinforcement learning problem, and the practicability is also verified in several continuous control benchmark problems.

---

> > ### Author Response · Authors · 2021-11-15
> > **Author Response to Reviewer cTPK (cont'd)**
> >
> > > However, the overall performance evaluations do not seem very competitive when compared to sota actor-critic algorithms, e.g. SAC can achieve a return over 10000 at 1e6, and it would be much better to include such algorithms for comparison.
> >
> > ZOAC unifies zeroth-order and first-order methods into an on-policy actor-critic architecture, aiming to preserve the advantages from both. Therefore, we mainly focus on comparison between on-policy RL algorithms due to their similarities. As far as we know, PPO[1] is one of the best performing and widely-used on-policy RL algorithms and therefore is chosen as the first-order baseline. On the other side, ES[2] and ARS[3] are two SOTA zeroth-order methods for RL problems (these two algorithms are included as representative of derivetive-free methods in popular RL libraries like Ray RLlib). Experimental results in our paper show that ZOAC outperforms all these baselines.
> >
> > Off-policy actor-critic algorithms like DDPG/TD3, and SAC that stem from Q-learning actually are a little bit different from on-policy actor-critic algorithms that stem from policy gradient, which leverage replay buffer and update the policy based on the learned $Q$-function. In general, on-policy algorithms have higher stability while off-policy ones have higher sample efficiency but may suffer from sensitivity to hyperparameter settings and brittle convergence.
> >
> > One of the chosen zeroth-order baseline, ARS, compared with DDPG and SAC in their original paper. In their results (Table 5 in Appendix A.2 in [3]), ARS outperforms DDPG in 3 out of 4 MuJoCo environments, while outperforming SAC in 2 out of 4. And in our paper, ZOAC outperforms ARS both in sample efficiency, final performance, variance over trials, and also robustness of the  learned policies. In our experiments, ARS performs consistently with the results listed in their own paper. Besides, one thing interesting is that, as two concurrent works, TD3 and SAC both reported very different results in their original paper: TD3 have the same or better performance than SAC in all 7 MuJoCo environments in TD3's paper (Figure 5 and Table 1 in [4]), while SAC outperforms TD3 in 4 out of 5 in SAC's paper (Figure 1 in [5]).
> >
> > We agree that a fine-tuned version of SOTA off-policy algorithms may achieve high sample efficiency with the usage of replay buffer. However, at present we mainly focus on integrating zeroth-order policy update into an on-policy actor-critic architecture, and therefore we only choose the most related methods as baselines in the paper. But we indeed consider to combine more improved techniques, including experience replay and other ZOO techniques, as described in the final section. It will be a challenging work combining them with ZOAC to further improve the performance while still maintaining the specific advantages which have been proven in this paper. We leave it as our future work and do not add SAC as one of the baselines at present considering the above reasons.

---

> > > ### Author Response · Authors · 2021-11-15
> > > **Author Response to Reviewer cTPK (cont'd 2)**
> > >
> > > > Actor with (64,64) hidden units is also not a standard choice, and it would be more interesting to see the results with different number of units, e.g. (128,128) and (256,256).
> > >
> > > We choose to use such a neural network mainly because two chosen baselines, PPO and ES, both use a (64, 64) network with tanh nonlinearities to produce their results in the original paper [1] [2], which is exactly the same as in ZOAC, and we believe that it will be a fair comparison using the same network architecture. Besides, a recent large-scale study on PPO shows that larger networks do not lead to higher performance. Instead, actor with (64, 64) hidden nodes outperforms larger networks with (256, 256) or (512, 512) hidden units in 4 out of 5 MuJoCo environments (Figure 18 in Appendix E.2 in [6]).
> > >
> > > However, we would like to thank you for pointing this out. We conduct additional experiments on the same environments with four different policies, from fewer parameters to more parameters:  Toeplitz matrix, matrix, network with (64, 64) units, network with (128, 128) units, and to see the performance of ZOAC when optimizing these policies. Toeplitz matrix is a kind of compact policies with parameter sharing schemes, each element depends only on the difference between the row index and the column index. A general dense matrix $M\in\mathbb{R}^{m\times n}$ has $m\times n$ parameters, while a Toeplitz matrix $T\in\mathbb{R}^{m\times n}$ has only $m+n-1$ parameters. We use the same set of hyperparameters for the first two policies, and another for the last two, and results show that a Toeplitz matrix is far inferior to a general dense matrix, while network with (64, 64) and (128, 128) performs quite similarly to each other. For detailed results, please refer Appendix A.5 in the updated manuscript.
> > >
> > > [1] Schulman, John, et al. "Proximal policy optimization algorithms." *arXiv preprint arXiv:1707.06347* (2017).
> > >
> > > [2] Salimans, Tim, et al. "Evolution strategies as a scalable alternative to reinforcement learning." *arXiv preprint arXiv:1703.03864* (2017).
> > >
> > > [3] Mania, Horia, Aurelia Guy, and Benjamin Recht. "Simple random search of static linear policies is competitive for reinforcement learning." *Proceedings of the 32nd International Conference on Neural Information Processing Systems*. 2018.
> > >
> > > [4] Fujimoto, Scott, Herke Hoof, and David Meger. "Addressing function approximation error in actor-critic methods." *International Conference on Machine Learning*. PMLR, 2018.
> > >
> > > [5] Haarnoja, Tuomas, et al. "Soft actor-critic algorithms and applications." *arXiv preprint arXiv:1812.05905* (2018).
> > >
> > > [6] Andrychowicz, Marcin, et al. "What matters for on-policy deep actor-critic methods? a large-scale study." *International Conference on Learning Representations*. 2020.

---

> > > > ### Author Response · Authors · 2021-11-15
> > > > **Author Response to Reviewer cTPK (cont'd 3)**
> > > >
> > > > We hope that our response has addressed your concerns, and we sincerely wish you can reconsider your recommendation for our work. If you still have other comments, or if we have misunderstood your concerns, please let us know and we would be happy to make further response or revision.

---

### Author Response · Authors · 2021-11-24
**General Response to All Reviewers**

We thank the reviewers for their thorough and insightful feedback. We are glad that the reviewers gave generally positive comments on our work:

1. Our idea and the proposed algorithm is novel (Reviewer cTPK), interesting (Reviewer eX28), well-motivated and elegant (Reviewer wk6f);

2. Experiment on several benchmarks give convincing and promising outcomes, which demonstrate the advantages of ZOAC, including sample efficiency and robustness of the learned policies (Reviewer cTPK, eX28, wk6f);

3. The manuscript is overall well-written and easy to understand (Reviewer wk6f);

4. The literature review is clear and comprehensive, which demonstrates the desirability and the technical contributions of our proposed algorithm (Reviewer eX28, wk6f);


We have addressed the concerns of all reviewers individually and have updated the manuscript accordingly as follows:

1. We have updated the schematic diagram of ZOAC (Fig 1a) to make it more clear and informative  (Reviewer wk6f);

2. We have performed additional experiments on two more policies to further show the wide adaptability of ZOAC, including a Toeplitz matrix with parameter sharing scheme and a larger neural network with (128, 128) hidden nodes. The results and discussions are attached in Appendix A.6 (Reviewer cTPK);

3. We have rewritten the "Robustness Comparison" paragraph in Section 4.1 to more clearly analyze the advantage of ZOAC on generalization and robustness. Further discussions and the results of robustness comparison in all tested environments are also added in Appendix A.5 (Reviewer eX28);

4. We have also more detailedly analyzed the underlying problem of two zeroth-order baselines in Section 4.1 that the exploration trick used in these methods (subtracting survival bonus in ARS, transforming accumulative rewards into rankings in ES) actually alter the objective function that is being optimized, which may lead to unexpected outcomes;


We address some common concerns as follows:

1. The main technical innovation of our work is that we unify evolution based zeroth-order methods and policy gradient based first-order methods into an on-policy actor-critic architecture to preserve both advantages. Under this architecture, we also perform a timestep-wise perturbation to encourage exploration in the parameter space and derive the corresponding zeroth-order policy gradient. The derivative-free nature makes it useful especially when gradient information is hard to obtain or even unavailable. However, the current paper focus on proposing the novel idea and practical algorithm, and therefore we still choose the common continuous control benchmarks as the testbed and apply it to two different types of policies to demonstrate its superiority. Applying ZOAC to specific RL problems (e.g., problems that involve non-differentiable components and therefore need derivative-free update, which are currently solved by conventional zeroth-order optimization methods) is beyond the scope of this paper, but will definitely be one of our future work;

2. The choice of baselines. Zeroth-order policy search has its own advantages and is more suitable for some specific kinds of RL problem, as demonstrated in our paper and also in our comments. Actually our initial motivation is to improve the efficiency and stability of these methods, and in this paper we accomplish this by nicely integrating it into a popular on-policy actor-critic architecture. We demonstrate the advantages over conventional zeroth-order policy search both theoretically and empirically. Besides, considering the similarities between ZOAC and gradient-based on-policy actor-critic methods, we also choose PPO (SOTA among this kind of methods) as one of our baselines, and experimental results shows that ZOAC actually outperforms PPO, which can be ascribed to the difference on the exploration way and also the objective function. We did not directly compare ZOAC with off-policy algorithms like SAC/TD3, because (1) a carefully tuned version of off-policy algorithms can achieve very high sample efficiency with the usage of replay buffer, which is also their biggest advantage and meanwhile the biggest limitation of zeroth-order methods, as described in the introduction section; (2) these methods actually stem from Q-learning and are quite different from either of the chosen baselines and ZOAC. Therefore, we only compare with the most related methods that has similar application scenarios. In the future, we indeed consider to further integrate more improved techniques from both sides (e.g., sampling tricks/low-dimensional manifold identifications from ZOO and experience replay/target networks from first-order RL) for further performance improvements.

Therefore, we think these concerns may not outweigh the overall technical contributions of our work.

Thanks again for reading our article carefully and giving very constructive suggestions.

---

### Decision · Program_Chairs · 2022-01-20

**Decision:**

Reject

**Comment:**

The paper proposes a new reinforcement learning actor-critic type algorithm for parameterized policy spaces. The actor builds gradient estimates derived from perturbations of the policy (in the spirit of simultaneous perturbation stochastic approximation (SPSA) or Flaxman-Kalai-McMahan's "Gradient Descent without a Gradient" idea), while the critic is based on standard temporal difference (TD) learning. The algorithm is benchmarked, along with other well-known techniques, on Mujuco-based environments where it is seen to often perform well.

There were several concerns raised by the reviews initially, including the validity of the value function obtained by the rather non-standard perturbation of the behavior policy suggested in the paper, the necessity of the zeroth order scheme, the impact of the hyperparameter N, the lack of clarity about the overall algorithmic flow, and the lack of more contemporary baselines such as SAC, A3C and TD3.

Most concerns appear to have been addressed by the author(s) in their detailed responses, and new explanations have been added with significant effort, to the credit of the author(s). While the paper breaks new ground in the conceptual sense, and the reviewers are borderline positive about the paper, I am afraid that parts of the paper, especially relating the the soundness of the algorithm, are still unclear and not concretely motivated. This, coupled with the low confidence levels expressed in the reviewers' evaluations, renders the paper's form too preliminary at this stage to merit acceptance.

For instance, I notice upon a careful reading of the paper the following issues:

(a) Equation (7) is derived by claiming that $V^\beta(s_t)$ is uncorrelated with the Gaussian noise $\epsilon$. However, I fail to see why this should hold, since the paper mentions, in the paragraph before equation (6) that $\beta = \pi_{\theta + \sigma \epsilon}$, so $\beta$ ostensibly clearly depends on $\epsilon$.

(b) The motivation behind the objective $J_{ZOAC}$ in (6), and the quantities involved in its definition, is rather opaque. For instance, the right side of (6) suggests an infinite horizon discounted reward criterion, whereas the expectation is taken with respect to $d^\beta$, the "stationary distribution" of the policy $\beta$. How/why is this justified? I would expect the use of the discounted occupancy measure here, instead of the (long term) stationary measure which washes out any near-term trajectory effects.

(c) The paper mentions that $\epsilon$ is a sequence of random perturbations *per time step* in (6) as opposed to the usual ES perturbation of a one-time perturbation. However, the size of the covariance matrix $I$ in (6) and (3) are not explicitly distinguished, leading to much confusion in the mind of the keen reader.

I hope that the author(s) can utilize the feedback from the reviews in order to put up a significantly clearer and solidly motivated paper in the next round, so that its conceptual merits can be proven without doubt. Thanks and best wishes.